# Application of Epidemiological Methods in a Large-Scale Cross-Sectional Study in 765 German Dairy Herds—Lessons Learned

**DOI:** 10.3390/ani14091385

**Published:** 2024-05-06

**Authors:** Roswitha Merle, Martina Hoedemaker, Gabriela Knubben-Schweizer, Moritz Metzner, Kerstin-Elisabeth Müller, Amely Campe

**Affiliations:** 1Institute of Veterinary Epidemiology and Biostatistics, School of Veterinary Medicine, Freie Universität Berlin, Königsweg 67, 14163 Berlin, Germany; 2Clinic for Cattle, University of Veterinary Medicine Hannover, Foundation, Bischofsholer Damm 15, 30559 Hannover, Germany; martina.hoedemaker@tiho-hannover.de; 3Clinic for Ruminants, Ludwig-Maximilians-Universität Munich, Sonnenstr. 16, 85764 Oberschleißheim, Germany; g.knubben@lmu.de (G.K.-S.); m.metzner@lmu.de (M.M.); 4Ruminant and Swine Clinic, School of Veterinary Medicine, Freie Universität Berlin, Königsweg 65, 14163 Berlin, Germany; kerstin-elisabeth.mueller@fu-berlin.de; 5Department of Biometry, Epidemiology and Information Processing (IBEI), WHO Collaborating Centre for Research and Training for Health at the Human-Animal-Environment Interface, University of Veterinary Medicine Hannover, Foundation, Buenteweg 2, 30559 Hannover, Germany; amely.campe@tiho-hannover.de

**Keywords:** good epidemiologic practice, dairy cows, cross-sectional study, incidences of chronic diseases, dairy farms

## Abstract

**Simple Summary:**

The “PraeRi” study, conducted by three German veterinary universities from 2016 to 2020, was aimed at enhancing dairy health and welfare. With 765 visited dairy farms and 101,307 animals examined, the designed study provided an opportunity for improving animal well-being. Researchers applied good epidemiologic practice and concepts proven in large-scale studies, in a study comprising three different regions in Germany with varying farm characteristics. A sample size of 250 farms per region, determined through stratified sampling based on farm size, ensured representative estimates. According to the information provided by the farmers, mastitis was the most frequently occurring disease in their herds (14.2% to 16.3% of the herd—depending on the region). For most disorders, prevalence data were lowest for the region South compared with the two remaining regions. Statistical analyses identified risk factors, with results communicated through individual reports and benchmarking flyers to participating farmers. Challenges arose from managing practical procedures and communication due to the project’s vast scale. Harmonizing data management and hypothesis testing across involved parties added complexity. Despite challenges, the PraeRi study considerably contributed to advancing dairy health and welfare practices.

**Abstract:**

From 2016 to 2020, the “PraeRi” study, conducted by three German veterinary universities, was aimed at enhancing animal health and welfare in dairy farms. With 765 dairy farms visited and 101,307 animals examined, this study provided a basis for improving animal health and welfare. The study population comprised three different regions representing a broad variety of characteristics. To ensure representative estimates, a sample size of 250 farms was determined for each region, employing a stratified sampling plan based on farm size. According to the information provided by the farmers, the most commonly occurring disease in their herds was mastitis without general disorder (14.2% to 16.3% of the herd—depending on the region). For most disorders, prevalence data were lowest for the region South compared with the two remaining regions. Multivariable regression analyses were performed to identify risk factors for various target variables, and the results were communicated through individual reports and benchmarking flyers to participating farmers. The authors encountered challenges in management and communication due to the project’s size in terms of personnel, data, and farms examined. Harmonizing data management and hypothesis testing across all involved parties added complexity.

## 1. Introduction

From 2016 to 2020, the large-scale study “PraeRi” (animal health, hygiene and biosecurity in German dairy herds—a prevalence study) was carried out by three German veterinary universities: University of Veterinary Medicine Hannover (North), Freie Universität Berlin (East), and Ludwig-Maximilians-Universität Munich (South). The overall goal of the study was to improve animal health and welfare in dairy farms. The main task was to provide reliable and representative estimates of basic dairy cow health indicators, such as mastitis, lameness, infectious diseases, etc., because prevalence estimates had not been available at that time. The latter data form an ideal base for strategies to improve animal health and welfare.

Three clinics and two epidemiologic institutes participated in the study. Eleven senior researchers and 43 study veterinarians worked on the project. In total, 765 dairy farms were visited, and data were collected (interviews with 376 questions; standardized measurements of stalls; examination of 186,160 animals; laboratory analyses) and recorded in 35 tables with 1522 variables in a relational SQL-database. The study was funded by the German Federal Ministry of Nutrition and Agriculture with a total amount of more than EUR 4.6 m (support codes: 2814HS006 (Hannover), 2814HS007 (Berlin), 2814HS008 (Munich)). The final report to the ministry consisted of a main document with 262 pages and eight appendices enclosing more than 276 pages (https://ibei.tiho-hannover.de/praeri/pages/69 (accessed on 31 March 2023)).

Researchers do not often have the chance to be part of a large-scale epidemiologic observational study because such studies are expensive and time-consuming. Only few funding bodies such as governmental research institutions are able and willing to invest large amounts of money into studies providing medium- and long-term results.

Large observational studies offer many possibilities to collect representative and reliable data on complex multifactorial issues. Dairy cow health disorders originate from complex interactions of housing conditions, herd management, and health management, as well as the farmers’ personality and characteristics of the individual animal. The collection of data that include a high number of farms and cows as well as many variables is a precondition for sound and reliable statistical modeling.

Good epidemiologic practice (GEP) is a guideline for the state-of-the-art conduction and analysis of epidemiologic studies [1]. Large-scale studies have a higher chance of applying these guidelines than smaller ones.

Large-scale studies, however, pose challenges that do not occur in smaller studies. Data collection on a large number of farms including plenty of variables requires good study-management. As an example: when various observers collect data, good interobserver agreement must be safeguarded by repeated trainings. In addition, the analysis of the data is not as straightforward as in smaller studies with one or few target variables and less hierarchical structures. An analysis strategy must be agreed on by all researchers involved, and future use of the data must be coordinated.

We use our experiences from the study “PraeRi” to present and discuss an approach on how GEP and standard epidemiologic methods can be successfully applied to a large-scale study. Since this study was an observational study, we followed the guidelines of the STROBEvet statement [2] to structure the following text.

## 2. Materials and Methods

We conducted a cross-sectional study addressing animal health, hygiene, and biosecurity to achieve representative and reliable prevalence estimates of animal health disorders in German dairy farms.

### 2.1. Study Population

Regionalization: Three structurally different regions were studied, all of which make a significant contribution to milk production in Germany, but were different concerning farm size distribution and farm management. The latter differences were already reported by one of the co-authors in 2012 [3].

#### 2.1.1. Sample Size of Farms per Region

Different target variables: We intended to calculate one (overall) sample size suitable for several different target variables. Hence, it was to be considered that each target variable had a different distribution (point estimate and standard deviation), when estimated on a farm level (i.e., distribution of milk yield per farm), or was measured as a percentage for the presence of a characteristic. As most of the target variables were originally collected on an animal level (i.e., quantitative value or dichotomous value for the presence or absence of a disease), these had to be aggregated on a farm level to obtain the statistical unit as the farm. For those target variables that were quantitative in nature, a measure of central tendency per farm was calculated. The distribution of this measure of central tendency was the target value to be interpreted in terms of content. For those target variables that were dichotomous in nature (e.g., presence of disease), the prevalence was calculated as the percentage of animals affected by the disease in question.

One for all sample size: To calculate the sample size, we took estimates from a previous study as a basis. First, we calculated sample sizes between 10 and 297 for quantitative target variables given a precision of 1 to 5 and a standard deviation of 7 at a confidence level of 0.95 to 0.99. Then, we calculated sample sizes between 19 and 2536 for dichotomous variables given a prevalence from 0.05 to 0.5 and a precision from 0.025 to 0.1. All calculations were conducted with NCSS PASS version 13.0.8 and under the consideration of Glaser and Kreienbrock [4]. Based on these scenarios and with feasibility in mind, a sample size of 250 farms was determined for each region.

#### 2.1.2. Farm Size as Stratifier

To prevent a biased study population with regard to farm sizes, a sampling scheme was developed in which stratification was made according to small, medium, and large farms (control of selection bias). Farm sizes were chosen as cut-offs for this categorization, which allowed the target population to be assigned to three equally sized groups. These cut-offs were determined individually for each study region and transferred to the study population. Consequently, 83 farms were to be investigated per farm size category and region.

#### 2.1.3. Sampling Procedure

A random sample of farms was drawn from the National Traceability and Information System for Animals “Herkunftssicherungs- und Informationssystem für Tiere” (HIT; https://www.hi-tier.de/ (accessed on 31 March 2023)) [5], with selection stratified by state or region and farm size. Based on the experiences of a previous study [6], a response rate of approximately 30 to 40% was anticipated.

For each region, 1250 farms were randomly selected, 5 times more than what was needed to cover a planned participation rate of at least 20%. Region North included the federal states of Schleswig-Holstein and Lower Saxony. Region East was represented by the states of Mecklenburg-Western Pomerania, Brandenburg, Saxony-Anhalt, and Thuringia, and Bavaria was studied for region South.

In Bavaria, the use of the HIT data was not approved by the Bavarian State Ministry for the Environment and Consumer Protection. Alternatively, the sampling could be performed using the database of the Milchprüfring Bayern e.V. (MPR; https://www.mpr-bayern.de/en (accessed on 31 March 2023)). Here, about 90% of the dairy farms located in Bavaria were registered.

Due to the low overall participation rate and the unintended inclusion of cattle farmers without dairy cows in the selection population, an in-depth analysis of the composition of the target population and the selection population from HIT was conducted in summer and early fall 2017. The objective was to (1) initiate a second draw of address data and, in doing so, (2) better reflect the true distribution of farm sizes of dairy farms. Therefore, the HIT animal data were also cross-checked with data from the federal state control associations (Dairy Herd Improvement associations (DHI), [7]). For this purpose, the September 2016 data were used to ensure comparability with the HIT data, which were also from September 2016.

Finally, a different type of address data management was defined for each region:Region North (N): The farm size classes were recalculated using the DHI information on farm sizes in Lower Saxony and Schleswig-Holstein and applied to the HIT data. The farm size information came from DHI data, while the address data drawing continued to be collected in HIT.Region East (E): Here, too, new farm size classes were determined using DHI data from Brandenburg, Saxony-Anhalt, Thuringia, and Mecklenburg-Western Pomerania. The address data were then also drawn from DHI data. A change in the address data source in the region East was justified by the fact that a very high proportion of dairy cow farmers (mean: 55%) and dairy cows (mean: 90%) were also members of a state control association in the states concerned. The advantage of writing exclusively to dairy cow farmers thus outweighed the disadvantage of not reaching those dairy cow farmers who were not members of the state control association. Due to the change in address data management in region East, it was unavoidable that farms were contacted twice with the second sampling. However, the DHI excluded those farms that had already participated in the study in advance.Region South (S): The calculation of the farm size classes was based on the data of the MPR. First and second data draws were made from the address data of the MPR, since access to address data from HIT had not been granted in Bavaria.

The finally realized farm size classes are shown in Table 1, and the final participation rate per region and class in Table 2 and Figure 1.

#### 2.1.4. Sampling of Animals per Farm

On large-scale farms of region East, it was not possible to examine all animals on a farm. The sample size for each farm was calculated to estimate an expected prevalence of 40% at a confidence level of 95%, with a power of 80% and a precision of ±5%. Thus, no more than 130 animals were sampled per farm in region South and no more than 213 in region North. In region East, all animals were sampled in farms with up to 159 animals, 166 in farms with 160 to 292 animals, and no more than 292 animals in farms with 293 animals or more [8]. The methods to ensure an unbiased selection of the cows evenly distributed over compartments were set out in the project’s SOP manual.

For the calves, a sample calculation per farm was also performed. Here, an expected prevalence of 40% was to be estimated at a confidence level of 95%, with a power of 80% and an accuracy of ±10%. Consequently, no more than 33 calves were examined in region South and no more than 54 in region North. In region East, all animals were sampled up to a farm size of 40, 40 in farms with 41 to 73 animals, and from 74 calves onwards exactly 73 calves were sampled.

### 2.2. Questionnaires and Survey Forms

Based on the examination catalog from a previous project, as well as on literature research and expert opinions, the questionnaires and survey forms for the herd examinations were created and validated. The process of variable selection and question formulation was based on pre-formulated hypotheses, which were specifically linked to the disorders and performance features to be investigated. The questionnaires were used to record the relevant disease and performance characteristics of dairy cows, calves, and young stock in Germany, as well as potential factors influencing them (Appendix A). The survey forms were used to investigate cows (e.g., body condition score [9], locomotion score [10,11], and hygiene score [12]). A new tool for the assessment of hygiene in calves was developed and published by Kellermann et al. [13].

It was determined by which method, from which source, at which level, and for which animal groups information should be collected. Feasibility, validity, and repeatability were considered. An internal and external validation of the questionnaires and survey forms was performed via testing in pilot farms (South: n_farms_ = 5, n_animals_ = 357; East: n_farms_ = 3, n_animals_ = 944; North: n_farms_ = 6, n_animals_ = 1424). Based on the experiences from the pilot phase, the questionnaires and survey forms were modified once again. In the end, 15 questionnaires and 15 survey forms were created, some of which had to be filled out only once, and some of which had to be filled out several times (e.g., for all different silages in use).

### 2.3. Database and Homepage

A homepage (www.PraeRi.de (accessed on 31 March 2023)) with information on the project, its objectives, the study teams, and the status was available for animal owners and veterinarians. It is still operated and can be used to look up results (including the final report).

An SQL database with a web interface was set up with the scripting language PHP and maintained on a virtual server at the University of Veterinary Medicine Hannover Foundation, Germany. There were different user groups, and the data input could be conducted and decentralized. All data of the project could be viewed by all users according to their user rights. Functions were available for general, as well as individualized, export of data. After the creation of the database and input interfaces, an internal (with colleagues from the IBEI) and external (with colleagues from study team North) check was carried out and any errors were corrected, or missing items added. The database comprised 35 main tables, 132 check tables, and 1522 data fields. It was used by about 40 users with different data access and user rights. In addition to the data from the questionnaires and survey forms, most of which had to be entered manually by the study veterinarians, import functions were available, which were also set up for data from external sources (e.g., HIT (www.hi-tier.de (accessed on 31 March 2023)), DHI, and LUFA Nord-West (accredited service laboratory of the Lower Saxony Chamber of Agriculture, https://www.lufa-nord-west.com/ (accessed on 31 March 2023)) [14], ration key figures, parasitology laboratory). Throughout the reporting period, various requests were raised by the study teams, errors were uncovered, and changes and additions were made. Therefore, various additional programming was carried out to facilitate the use of the database, data entry and data correction. Furthermore, explanations on the use of various database parts were available (user manual).

### 2.4. Farm Recruitment

Employees of the HIT database created a sample according to the above-mentioned sampling requirements. The sample consisted of an address list and was sent to the top state representatives of the veterinary authorities to preserve data protection. They passed it on to the study units responsible for address data management. In the region East, this was executed by the regional state control associations for milk testing. The latter institutions ensured the address data were administered and established contacts with the farmers asking if they were willing to participate. In Schleswig-Holstein, this was carried out by the Ministry for Energy Transition, Agriculture, Environment and Rural Areas, and in Lower Saxony by the project secretary of the Clinic for Cattle at the University of Veterinary Medicine Hannover. Since the Bavarian State Ministry for the Environment and Consumer Protection did not agree to the use of HIT data for farm recruitment in Bavaria due to data protection reasons, the random sampling and printing of the cover letters (see below) in region South was carried out at the MPR.

Uniform cover letters were prepared for the participants in the three regions. The selected farms received such a cover letter by postal mail together with study-specific information (in the form of a flyer), a reply postcard, and a postage-paid envelope. Persons who wished to participate in the study returned the reply postcard to the respective regional office (by postal mail, fax, or e-mail).

When receiving the farmers’ positive reply, the study veterinarians gained access to the contact data for the first time. They made an appointment for a telephone interview, in which the contents of the study and the investigations planned on the farm were explained to the farm managers. Farms with farm-gate sales and farms which delivered their produced milk to neighboring countries were not included. Then, they requested information required to prepare adequately for the farm visit. At the end of the telephone interview, an appointment was made for the visit. Farmers were also informed that they must ensure access to their HIT- and DHI-databases during the farm visit, so that the previous year’s data could be downloaded.

### 2.5. Quality Assurance Measures

Several measures were taken to ensure that data were collected consistently across all three study regions, to the extent possible.

#### 2.5.1. Sampling of Animals per Farm

Through the intensive exchange during the development of the questionnaires and survey forms, all study veterinarians benefited from the different experiences that the individuals contributed to the project. At the same time, the literature research and the exchange with additional experts led to a familiarization with the different topics.

#### 2.5.2. Training to Enhance Interobserver Reliability (IOR)

For quality assurance, the study veterinarians were introduced and trained in livestock surveys (three-day training in September 2016).

After the completion of the pilot phase of the project, a one-day alignment of different measurements took place in Hannover at the university’s Teaching and Research Farm in 2016. The aim of the comparison was to determine whether and to what extent the study veterinarians (observers) agreed in their assessment of the animals. This was conducted to minimize the possible effects of individual observers and thus to be able to collect the PraeRi data in a harmonized way. The criteria for locomotion and hygiene assessment, as well as the measurement of cow sizes were fulfilled. To identify substantial deviations in single observers, an exclusion test was performed according to Ruddat et al. [15]. In total, there was already good to very good agreement for some measurements at the beginning of the study. For recalibration of the study veterinarians, two further IOR comparisons took place. Systematic regional effects could not be identified in this observer assessment.

At the second training in 2017, 14 calves and 60 cows were evaluated by each of the 16 study veterinarians present. The third training was held in Munich in 2018. During this two-day training, 60 cows and 20 calves were evaluated again by the 15 study veterinarians present. Additionally, 13 cows in tethered housing were assessed. Furthermore, photos or videos were evaluated and discussed jointly by all study veterinarians in the context of a group discussion for distinct measurements, which had emerged as problematic in the previous training sessions.

The data of the assessment of the cows and calves were analyzed, and the results were communicated in a general report to all project partners. In addition, observers whose results significantly deviated from the majority of study veterinarians with respect to at least one observation received a personal notification which enabled them to adjust their assessment in the future. An analysis of the data from the second and third comparison showed that there was an overall satisfactory agreement between the study veterinarians. Except for differences in the auscultation of the calves, no evidence of region-specific differences could be found. Due to partly only very small differences between the examined animals, the agreement between the study veterinarians could not be evaluated with the desired certainty for all characteristics.

#### 2.5.3. SOP Manual

To ensure a uniform procedure for the collection of data by the study veterinarians, standard operating procedures (SOPs) were developed and summarized in a collection of methods (SOP manual). In addition to the SOPs for the individual questionnaires, the manual also contained a glossary to explain the special terms used to the farmers and standard procedures for filling out the questionnaires. Any ambiguities that arose during the study regarding the collection of data were discussed in regular telephone conferences of the study veterinarians and corresponding decisions were documented in the manual.

#### 2.5.4. Leaders Video Conferences and Collaborative Meetings

Video conferences of the leaders: Video conferences of the project managers of the three study teams took place in monthly intervals. These were used to exchange information on the current status of the studies, to discuss and solve problems that had arisen, and to agree on the next steps in the project.

Consortium meetings and coordination of the analyses: a consortium meeting was held in Berlin in 2018 to coordinate the work in the context of data analysis, of the preparation of the final report, and the scientific publications of the overall results. Eight thematic working groups were formed that prepared detailed descriptive evaluations and the appropriate statistical models (see below for more information on the approach). Furthermore, a two-day collaborative meeting for the actual analyses took place in Hannover in 2019, where the two epidemiological institutions presented detailed instructions for the steps of the analyses.

Telephone conferences of the study veterinarians: In addition, telephone conferences between the study veterinarians and the epidemiological institutes were held at approximately bi-weekly intervals. In these, mainly emerging issues or peculiarities observed on the farms were discussed and a decision was made on how to handle such cases. Questions that remained unresolved were discussed and, if necessary, final decisions were made in the monthly leaders’ video conference. All decisions made during telephone and video conferences were subsequently included in the manual.

### 2.6. Data Analysis

Data were pseudonymized, i.e., the identification patterns such as the name and address of the farm were stored separately. The data set used for analyses did not contain any data that would enable third-party member to identify the farms. The list for re-pseudonymization was only accessible to the regional study teams in order to send feedback to the farmers.

#### 2.6.1. Feedback Letters to Farmers

In total, two feedbacks were sent to the participating farmers. The first letter represented an individual report that was prepared a short time after the farm visit and included an overview of all results of the investigations: It included the evaluations of the skin lesions as well as of the hygienic conditions, a number of tables and figures displaying the occurrence of lameness, the distribution of the body condition score, and an overview of the calf weight related to age. An example of this letter is given in Appendix A.

After total completion of the data collection process and baseline descriptive analyses, another round of feedback was sent to all participating farmers in the form of a benchmarking flyer, in which individual farm data were compared with the distribution of data of all study participants (see Appendix A).

#### 2.6.2. Plausibility Controls

The epidemiology groups performed general plausibility checks on approximately 230 quantitative variables from the interview questionnaires and approximately 50 variables from the data entry forms. For these variables, no database internal validations during data entry were applied, which would have caused error messages in the case of non-plausible entries. Therefore, the observers went through a number of lists with potentially implausible values and checked questionnaires and survey forms for the correct values. Accordingly, approximately 130 related questions (so-called filter questions) were checked for logical relationships. When these respective data were entered into the database, a variable-specific, logic-based pre-assignment of dependent answers initially took place automatically. However, this could be changed by the data entry person if needed, which made it necessary to check the data records after data entry. Following the plausibility checks, database queries were defined and programmed to allow the study teams to retrieve clean and summarized data sets.

In addition, database queries were created that related to specific questions. On the one hand, these were adapted from the planned analyses in the working groups. On the other hand, imported data from the external data sources HIT and DHI had to be checked very intensively. Some information was included in both data sets. The information, however, included was not always consistent. For example, information on the lactation number of animals did not match the recorded number of calvings. The reasons for such discrepancies (here, for example, the non-reporting of stillbirths in HIT) were clarified. We identified which data source had the greatest completeness and reliability for the respective information. With the help of contact persons at HIT and “Vereinigte Informationssysteme Tierhaltung w.V.” (https://www.vit.de/en/ (accessed on 31 March 2023)), discrepancies could be traced, and further work could be conducted with these findings in the best possible way.

All plausibility checks required intensive exchange between the study teams. Expertise and experience from previous farm visits were considered, as well as region-specific features. For the specific plausibility checks, the epidemiologists sent lists of data sets to the study teams, which then checked these and made necessary corrections in the database. All plausibility checks resulted in a very low number of missing values, on the one hand, and an extremely high quality of the available data, on the other hand.

#### 2.6.3. Multivariable Regression Analyses

Working groups: As described above, topic-specific working groups were formed to guide the inductive statistical analyses. Each group consisted of study veterinarians from all three regions, an epidemiologist and a project leader.

The following seven topics were addressed in this way:Udder Health;Reproduction;Metabolism;Limb health and lameness;Technopathies such as skin lesions;Calves/young cattle;Feeding;Infectious diseases/biosecurity.

Communication within the working groups took place via telephone (bilateral/conference), video conference, or e-mail. In addition, internal working group meetings were held at one of the study sites on an irregular basis. The task of the individual working groups was to define target values in their area and to assign potential risk factors. Hypotheses: A hypothesis-based approach was chosen to select the variables for each model. First, hypotheses were formulated including one target variable (outcome) and one or several influencing factors. The whole consortium agreed on a summarized list of hypotheses.

In the next step, causal diagrams were created with the help of a hypothesis list. After identifying the target variable, a causal directed acyclic graph (DAG) (http://www.dagitty.net/ (accessed on 25.05.2023)) [16] was created, including all known and expected influence factors on this outcome, from existing and/or expert knowledge. Drawing arrows between the variables enabled the scientists to identify confounders (common causes for influence factor and outcome) as well as variables causing selection or measurement bias when analyzing the effect of one specific variable on the outcome. An example of a causal diagram is shown below (Figure 2). For the statistical analyses, some of the influence factors were latent and were, therefore, represented by observed variables in the data collected.

##### Example Clinical udder Infections

Hypothesis: Cubicle cleanliness has an influence on the incidence of clinical udder infections in such a way that dirty cubicles negatively affect udder health.

Target: Frequency of clinical udder infections, e.g., in the last 12 months before the farm visit related to the number of dairy cows (lactating and dry cows) in this period.

Influence variable: “Cleanliness of cubicles”:Step 1: Define the animal group under consideration, e.g., lactating.Step 2: Definition of the hierarchical level, e.g., at the plant level, at least one compartment with the grade “soiled” [yes/no].Step 3: Description of influencing variable, e.g., farm has at least one dirty compartment with lactating dairy cows yes/no.

In most cases, the statistical unit was the farm, but the data were sometimes also collected at the animal, compartment, barn, and possibly site level. Data agglomeration at the various levels was therefore necessary, i.e., “percentage of cows with specific characteristics” (e.g., lameness) or “predominant category present in at least 80% of the cows” (e.g., breed). It became clear that the preparation of prudent sets of variables needed the expertise of both the study veterinarians and the epidemiologists.

#### 2.6.4. Statistical Analyses

Data from different levels of organization from one farm, such as an animal, stable, and farm, were linked via the farm ID.

Descriptive statistics: For qualitative variables, both the absolute number and the percentage of study units (farms, compartments, animals...) per expression (category and response option) were presented. This was conducted separately by the study region. Quantitative variables were considered based on their distributions (measures of central tendency and variation). Variables at lower hierarchical levels (e.g., barn, compartment, or animal level) were often analyzed at their level of origin as well as at the farm level. For the presentation of qualitative variables at the farm level, the percentage of each characteristic per farm was calculated. The distribution of these percentages was summarized per region. For quantitative variables, this distribution was calculated for an arithmetic mean and median.

The mean percentage of cows leaving the farm was defined as the percentage of lost animals out of all animals on the farm in the year before the farm visit. The culling rate considered the days an animal was present during the study period and provided additional information about the “rate” at which an animal was likely to leave the farm as a result of sale for breeding or slaughter. For example, a mean culling rate of 52.2 per 100 cows per year means that the probability of a cow leaving after a full year is 52.2%, or, in other words, that on average each cow will leave within about 700 days. The loss rate was calculated as
number per 100 cows and year = (number of lost animals/number of animal days in the period)∗365∗100 (1)

Information on reasons for leaving was obtained from the DHI and HIT data. From the DHI data, voluntary and involuntary reasons for cows leaving the farm were derived. When evaluating the figures, it must be considered that in a substantial number of cases, not single but multiple reasons led to the decision that the animal had to leave the farm.

The mortality rate was calculated as
number per 100 cows per year = Number of animals died/killed/number of animal days in the period)∗365∗100 (2)

Regression models: Generalized regression models were performed for each target variable to identify relevant risk factors for the target variables of interest (e.g., disease frequencies and skin lesions). If the target was on any other than farm level, a hierarchical model including the farm, and, if necessary, barn level as random factors was developed.

During data management, all variables required for the respective model were compiled. The documentation of the data management was conducted in an Excel file containing information on variable labels, data type, coding, and clear names of the variables, as well as a reference to the question in the questionnaire. The members of the working groups jointly determined which variables would be transformed (e.g., take the logarithm if there was no normal distribution) or recategorized (combine categories). Missing data were labeled with special missing values such as −77 (not applicable), −88 (do not know), and −99 (not specified).

First, a univariable generalized regression model was created separately for each influencing factor (univariable evaluations). In addition, a bivariate correlation analysis of all variables among each other was performed. The result of these steps resulted in the selection of the influencing variables which were potentially included in the multivariable model:Potential association with target variable, expressed as○Correlation coefficient > 0.1; or○*p*-value < 0.2.Factors indispensable to the content.

If two influencing variables were highly correlated, the one with the lowest *p*-value was selected or the two variables were combined into one.

##### Multivariable Analyses

The confounder analysis was performed separately for each of the influencing factors. A causal diagram was now created using these variables. In our example, the new infection rate of dry cows is the target variable, and the use of antibiotics during the dry period is the influencing variable (Figure 2). General dry cow procedures and breed can influence both the target and influence variable. Therefore, these were considered confounders and had to be included in the model, if the actual influence on the target variable was to be investigated. Straw bedding of the dry cows, on the other hand, only influenced the target variable.

Following the respective DAG, the influencing factors including all confounding variables and those that represented sources of selection and measurement bias were included in one (hierarchical) generalized regression model which served as the final model for the relationship between one influencing variable and the outcome. This resulted in several models for one outcome.

Since this procedure is very time-consuming, we had to use a more pragmatic approach for some outcomes: First, a maximum model was created in which all influencing variables were included with their associated confounders. Then, the variables were selected stepwise backwards by removing the variable with the largest *p*-value or with the smallest loglikelihood (-2ll). This was continued until only variables with *p* < 0.2 were left in the model. However, the variables that were indispensable always remained in the model. In the next step, the respective confounders for these variables were reinserted. Finally, possible interactions of influencing factors were determined and tested. Only interactions with *p* < 0.05 remained in the model. The final model was, therefore, composed of

indispensable factors in terms of content;factors with *p* < 0.2;their confounders from the causal diagram;interactions between two factors with *p* < 0.05.

Analyses were performed with SAS^®^ Software, version 9.4 of the SAS system for Microsoft (SAS Institute Inc. 2019, Cary, NC, USA). We used Proc Tabulate, Proc Means, Proc Univariate, and Proc Box-plot for descriptive analyses, as well as Proc Reg, Proc GLM, and Proc Glimmix for the regression models.

## 3. Results

### 3.1. Study Population

Participation rate: Overall, a participation rate of 6–9% was achieved (with slight regional differences) (Figure 1).

Representativeness of the study population: In this study, data were collected from 765 farms, and 86,304 individual cows, 15,003 individual calves, and 84,853 young animals kept in groups were examined (Table 3). The greatest number of animals was examined in the region East, which was due to the fact that dairy farming in the latter region—in contrast to regions North and South—is characterized by larger dairy operations with bigger herd sizes.. In addition, data from HIT and the milk performance tests were downloaded and evaluated. Table 4 presents the numbers of farms and cows in Germany compared to the figures in the PraeRi study. Due to the different numbers of farms and cows per region, the percentage of farms and animals included in the study was considerably higher (11 and 15%, resp.) in region East than in North (2%) and South (1%).

### 3.2. Baseline Results

#### 3.2.1. Study Population

A total of 765 farms (North: n = 253; East: n = 252; South: n = 260), roughly evenly distributed among the three farm size categories, were visited. The farm visits took place from early December 2016 to late July 2019. As outlined in Section 2, cows were evaluated using various grading schemes, and additional clinical examinations were performed on calves.

With respect to farm size, the mean number of lactating cows per farm was (minimum-maximum) 120 (North: 15–1165, n = 242), 396 (East: 3–3365, n = 249), and 51 (South: 6–254, n = 232), respectively. The mean number of dry cows was 17 (North: 1–225, n = 241), 67 (East: 1–492, n = 245), and 8 (South: 1–47, n = 228), respectively. It should be noted that only farms that participated in the DHI testing were considered here.

In all three study regions, the cows in the first lactation accounted for the largest proportion at approximately 30 to 35% per farm, followed by lactation numbers in descending order. Cows had a median lactation number of 2.6 (North), 2.5 (East), and 2.8 (South) lactations, respectively. Compared to the regions North and East, it is noticeable that cows in the region South potentially reached more lactations.

Of the cattle breeds listed in Appendix 6 of the Livestock Trade Regulations (2016), 23 were used as dairy cows on the farms visited, although some were beef cattle breeds. In the regions North and East, the German Holstein breed was the most common, with an average of 82.9% (Median: 95.0, SD: 28.0) and 84.1% (Median: 93.8, SD: 24.7) per farm, respectively, while in the region South, the Simmental breed was the most common, with an average of 80.4% per farm (Median: 100.0, SD: 35.1). In the region North, the order of other cattle breeds was Red Holsteins (Median: 2.1), cross-bred dairy cattle x dairy cattle (XMM), crossbred beef cattle x dairy cattle (XFM), and dual-purpose Red Holsteins. In the region East, the order of the other cattle breeds was XMM (Median: 3.9), Red Holsteins, other breeds, and Simmentals with also minor frequencies. In the South, the second most common cattle breed was Brown Swiss (Mean: 11.8%, Median: 0.0) followed by Holstein-Friesian (Mean: 3.9, Median: 0.0), other breeds, and XFM.

The proportion of farms participating in the DHI testing was very high at 90% and above, depending on the region. Compared to the region North (4.3%, n = 11) and the region East (1.2%, n = 3), the proportion of farms that did not participate in DHI testing was highest in the region South with 10.8% (n = 28). The mean annual milk yield (minimum–maximum) based on the DHI farm results from the last available audit year varied between 7606 kg (3712–10,598 kg, n = 231) in the South, 9055 kg (3597–11,927 kg, n = 241) in the region North, and 9250 kg (2739–12,907 kg, n = 249) in the region East, respectively.

#### 3.2.2. Farm Structure

Almost all farms in the regions North and East were run on a full-time basis (North: 98.9%, n = 250; East: 97.2%, n = 245). In contrast, 20.4% (n = 53) in the region South were part-time farmers. More than 80% of the farms were conventionally farmed. There were more organic farms in the South than in the other two regions (North: 4.4%, n = 11; East: 9.1%, n = 23; South: 13.9%, n = 36). In the region South, six farms (2.3%) reported being in the process of converting from conventional to organic. In more than 80% of the farms, cows were milked with conventional milking systems. A milking robot was present in 19.9% of farms in the region North (n = 50), in 14.7% of farms in the region East (n = 37), and in 11.9% of farms in the region South (n = 31).

During the farm visit, different survey sheets were completed for loose housing and tethering, depending on the housing system. Initially, it was not considered whether a certain type of housing was the predominant type of housing. However, based on the number of cows evaluated, it was possible to find out which was the predominant housing system, i.e., in which more than 80% of the cows graded were kept on the day of the farm visit. Loose housing was the predominant housing system in regions North and East (North: 92.9% of farms, n = 235; East: 96.0%, n = 242). However, this was the case in only 61.2% of the farms in the region South (n = 159). Overall, the tethering of cows was considerably less common than loose housing, but was more frequent in the South compared with the regions North and East (North: 3.6% of farms, n = 9; East: 1.2%, n = 3; South: 29.2%, n = 77).

The farmed area was significantly larger in the region East (1059.1 ha on average) than in the North (106.2 ha) and South (49.8 ha) regions. While about half of the farmland in the regions North and South consisted of grassland, this was true for 20% of the farms in the region East.

In addition to milk production, most farmers reared their own youngstock (Table 5). In the regions North and East, approximately one-third of the farms kept mating bulls and/or fattening bulls. Another more frequent cattle-related farm activity in the region East was suckler cow husbandry. In contrast, calves and young cattle from other farms were raised rarely in general, however, more frequently in the region North than in the other two regions.

Summer grazing varied in different animal groups and depending on the region and farm size. Comparing regions, animals of all age/lactation stages in the region North were on pasture in summer on many farms (North: approx. 60% of farms, East: approx. 20%; South: approx. 30%). With increasing farm size, the offer for grazing decreased. In all three regions, grazing very often was provided to dry cows (North: 71.9%, n = 182; East: 56.0%, n = 141; South: 35.8%, n = 93).

#### 3.2.3. Information on the Interview Partners

In the regions North and South, interview partners were owners and managers (North: 70.8%, n = 179; South: 81.2%, n = 211), predominantly. Equal partners were the second most interviewed group (North: 21.0%, n = 53; South: 11.9%, n = 31). In the region East, owners and managers (48.8%, n = 123) as well as herd managers and employees (42.5%, n = 107) were interviewed in equal numbers.

While the most common levels of education of the interviewees in the North were training as a master farmer (60.1%, n = 152) and training as a farmer (19.8%, n = 50), these training statuses were present equally often in the region South (master: 36.7%, n = 94; farmer training: 37.7%, n = 98). In contrast, the most common level of education in the region East was agricultural studies (59.9%, n = 150), followed by master farmer (17.1%, n = 43) and training as a farmer (14.3%, n = 36). It should be noted that, frequently, several levels of education were completed. The numbers given above, however, represent the highest level of education in case several levels were achieved. If we look at the level of training as a function of farm size, we notice a correlation in the regions North and East with higher levels of agricultural training, especially with a university degree. In the region South, where only a few interviewees had a university degree, a dependency on the size of the farm, although not as clear as in the other two regions, was only the case for training as a farmer and the acquisition of the title of master farmer.

#### 3.2.4. Animal Health and Animal Health Management

Integrated veterinary herd management (IVHM) was applied by 54.1% (n = 137) of farms in the region North, 59.9% (n = 151) of farms in the East, and 18.1% (n = 47) of farms in the South. On average, IVHM was carried out in two to three different areas per farm. In all three regions, IVHM was most frequently implemented in the areas fertility (North: 84.7% of the farms applying IVHM, n = 116; East: 92.7%, n = 140; South: 85.1%, n = 40), and udder health (North: 51.1%, n = 70; East: 83.4%, n = 126; South: 42.6%, n = 20). In the North, control of lameness, young stock health, and nutrition were further areas in which IVHM was applied by one-third of the IVHM farms, respectively. In the region East, lameness control (55.0%, n = 83) and youngstock health (71.5%, n = 108) were IVHM areas in even more farms, while, especially, lameness control (6.4%, n = 3) was not a common IVHM area in the South as in the other regions.

The mean incidence of diseases is shown in Table 6. It must be noticed that these incidences were based on information provided by the animal owners. Some of the figures came from very precise farm documentation, but in many cases only from an estimate by the animal owners (approx. 50–70% of the animal owners depending on the disease). All prevalence distributions had a right-skewed distribution. According to the farmers’ specifications, the most common disease in the herds was mastitis without general disorder (14.2% to 16.3% of the herd—depending on the region). For most diseases, prevalence data were lowest for the region South compared with the other two regions. Interestingly, in regard to foreign-body disease, mastitis with general disorder, and abortion, prevalence estimations did not differ considerably between regions.

Figure 3 shows that infertility, udder diseases, and disorders of the locomotor system were the main reasons for culling. While in the region North, the proportion of animals sold for breeding was significantly higher than in the other two regions, the South showed a high proportion of animals that left due to old age. In the East, on the other hand, a higher proportion of animals left due to underperformance than in the other two regions. In all regions, the high proportion of cows for which “miscellaneous” was recorded as the reason for culling was striking.

The culling risk was similar in all three regions (North: Mean: 33.5, Median: 32.9, SD: 6.9; East: Mean: 37.5, Median: 35.9, SD: 9.5; South: Mean: 37.7, Median: 36.3, SD: 9.4). The mean culling rate in the region North (Mean: 52.2, Median: 49.5, SD: 18.2) was slightly lower than in the East (Mean: 64.9, Median: 55.6, SD: 32.9) and in region South (Mean: 65.7, Median: 57.3, SD: 31.3).

The mean percentage for mortality, i.e., the percentage of dead/killed animals out of all animals on the farm in the year prior to the farm visit, was 3.7% (North: median: 3.2, SD: 2.5), 4.2% (East: median: 3.9, SD: 2.2), and 2.3% (South: median: 1.8, SD: 2.1). In the region East, the median mortality rate was 6.3, 4.7 in the North, and, in the South, it was 3.4.

## 4. Discussion

To the authors’ knowledge, a large-scale study, as reported in the present paper, has not been conducted on dairy cow health before. While the management of practical procedures and communication were challenging due to the sheer size of the project in terms of the amount of personnel, data collected, and farms and animals examined, we had to organize harmonized approaches for data management and hypothesis testing for all involved parties.

### 4.1. Sample Size

To receive valid estimates, the sample must be large enough and representative [17]. To find an optimal number of farms to be sampled, we started with calculating the sample size for several different target variables. This led to a multitude of different possible sample sizes, which was not satisfying. Thus, we needed to find a compromise between statistical necessity and feasibility in the field given the available time and personnel. From an epidemiological point of view, it must be concluded that estimates for some target variables were limited from this type of sampling compromise. However, this has merely manifested in a lack of precision in estimation. As all project partners were informed about this, it can be supposed that an interpretation of these measures was conducted with necessary caution.

Sampling type: We discussed two different approaches on the basis of which one can calculate the sample size (farm or animal as statistical unit). If one were to choose the animal as the statistical unit, one could then make a size-proportional selection of the farms. In this approach, large farms would have a higher probability of being studied than smaller ones because there are more animals there. This approach may initially seem more attractive for the study, as it requires fewer small farms to be studied than the approach on a farm level. This type of selection, however, has far-reaching consequences for subsequent statistical analysis. For descriptive analyses, each farm must be weighted according to its selection probability [17]. For inductive analyses, in which the possible factors influencing the target variables are examined, there are as yet, in some cases, no analysis methods at all with which this weighting can be considered. In particular, if a target variable (i.e., a health outcome) is categorical and, consequently, a logistic regression would have to be performed, there are no analysis methods available so far for this. Overall, therefore, this second type of selection would shift the effort from the phase of investigation to the phase of evaluation, and it would prevent evaluation in parts. Hence, we decided on a multistage sampling of farms and animals within farms. For us, this was the best compromise between good epidemiologic practice, on the one hand, and the comprehensibility of results and feasibility for later use by interested scientists or other parties, on the other hand.

Impact of region: Whether and what kind of regional differences occurred were not the main focuses of this study. Rather, the aim was to investigate a sufficient number of farms for each region. Our decision to divide the study into three regional studies had several advantages, but also disadvantages. It allowed us to adapt the definition of small, medium-sized, and large farms to be region-specific and thus to ensure that all farm sizes were covered by the study sample. However, the general population differed widely between the regions, from only 2256 farms in the region East to more than 32,000 farms in the South. For this reason, the percentage of farms represented in the study varied, leading to a different precision of the estimates. These structural differences also explain why it was possible to study 15% of the target population in the region East with 252 farms, whereas only 2% and 1%, respectively, could be covered with the same number of farms in the regions North and South.

For the estimation of the prevalence rates, this was not a big problem because the number of 250 farms was large enough even for the South to ensure a precision of ± 6% in the case of a 50% prevalence. For regression models and other statistical applications, this discrepancy became important because the percentage of participating farms in the East was higher than in the North or South, and thus the estimation was more precise. Even stratification does not solve this problem, so each model was run separately for each region [17]. In addition, one of the study’s objectives was to provide guidance to farmers concerning improvements in animal health and welfare. Due to structural differences, these recommendations needed to be region-specific, and thus data from each region needed to be reliable.

### 4.2. Representativeness of the Study Population

Source population: To ensure representativeness in terms of farm size, it was important to ensure that the composition of the farms studied reflected the composition of the target population as closely as possible. This was achieved very well in terms of the participating federal states. However, the average size of the PraeRi farms was higher than the data provided by the Stat. Bundesamt (region North: +17%; East: +35%; South: +19%) [18]. In order to keep this effect as low as possible, a sampling plan ensured that 1/3 of the farms per region were small. To classify this apparent discrepancy, it must be pointed out here that—as we described earlier—the farm sizes of dairy cow farms were biased downward in some databases. It should also be noted that the most recent data from the Stat. Bundesamt were from 2016, while our data collection took place until 2019. From this time, it is known that the average farm size had noticeably increased in all regions of Germany due to closures of small farms as well as restocking in the remaining farms (Landwirtschaftskammer Niedersachsen 2019; LKV Berlin-Brandenburg 2020). Comparing the PraeRi study with current reports of the DHI or similar, the PraeRi study represented the composition of the target population in terms of farm sizes well to very well. It can only be speculated as to whether the study results are transferable to other countries or not. We believe that if herd size, husbandry conditions (grazing, stable type), breed, etc., are considered, and also the climate is comparable as, e.g., in the neighboring countries, the study results will be valid to a certain degree.

Already during the preparation of the address data extraction from HIT it became clear that a pure separation of farms keeping dairy cows and other cattle farmers could not be ensured. This is because the information on whether a farm keeps dairy cows is only optional in HIT. For this reason, all cattle farmers who certainly or possibly kept dairy cows were defined as the source population. Before drawing the random selection, it was not possible to determine exactly what proportion of cattle keepers without dairy cows was in the selected sample. Thus, verification of the representative composition of the study population based on farm sizes was not possible without error. In addition to the mixing with suckler cow farms, it also had to be noted that some data in HIT were outdated and HIT listed farms that no longer existed. Since this apparently affected small farms more often than large ones, it can be assumed that the initially selected farm size cut-offs were somewhat too low. This example highlights that even when using an official register, such as the German HIT, data may be outdated or missing, and even mistakes must be taken into consideration.

As the sampling population contained cattle farmers without dairy cows, it must be assumed that the cut-offs determined to balance the sampling plan for farm size were also biased (assumption: the average farm size was slightly underestimated). In addition, the consequence that the source populations from which potential participants were sampled differed between the regions must be regarded as a possible source of selection bias.

Participation rate: The unexpectedly low participation rate required measures to meet the necessary sample size and to cover all categories of the sampling plan sufficiently.

A telephone non-response analysis in region North (n = 20 farms) confirmed that 14 of the farms surveyed did not keep dairy cows or that they planned to give or already had given up the farm. Measures such as tracking and interviewing owners of small farms as well as explicitly addressing small farms in the following letters, both seemed to have had an effect.

After initial difficulties in recruiting small (or, in Bavaria, medium-sized) farms in particular, a separate call for small farms was enclosed with letters of resampling. The importance of the participation of small farms for the significance of the study, and thus the livestock farmers themselves, was emphasized. This improved the participation rate.

The second draw of address data including the adjustment of the farm size categories yielded a full coverage of the sampling plan. Nevertheless, the overall participation rate was very low, at <10% across all regions and all farm size categories, which was much lower than in previous studies. Reasons for this may well be due to a heavy workload in agriculture rather than the scale of data collection in this study, as farmers made the decision to not participate at a time when they were not yet aware of the extent of data collection. In addition, studies with a focus on a specific disease or management problem might be more interesting compared to this rather general topic.

### 4.3. Prevalence Estimation

The aim of the study was to determine representative and reliable prevalence estimates for the most important dairy cow health parameters. This is because, to our knowledge, such large, time- and cost-consuming studies have been conducted very rarely, if ever, up to now. Alongside this, we were able to provide a data set that allowed for risk factor analyses, as well. The high completeness of the data set can be attributed to the efforts that have been taken by the plausibility checks and the following correction of implausible values whenever possible. Missing values are always a challenge for data analyses and the quality of a data set is also determined by the number of missing values.

This study is an epidemiological study, not an experiment. The study type is referred to as a cross-sectional study and follows specific rules for planning, implementation, and evaluation. It differs from other epidemiological studies in that, among other things, no control group is used here, as no scientifically selected groups are compared, but the status quo in the existing population is surveyed.

A good estimate of disease prevalence rates forms the basis for many studies—observational as well as experimental ones. This information is needed for the calculation of an adequate sample size in future studies that, e.g., want to investigate causal associations. Sample size calculation is necessary in the context of animal experiments. A good knowledge of inner-herd prevalence rates is also necessary to assume the potential improvement of a specific intervention measure. Thus, reliable estimates of prevalence rates in populations are a very important piece of information for current and future research and can thus not be underestimated [19].

Due to the higher number of observers and the fact that they worked at three different universities with possibly different opinions regarding the scoring and severity of lesions and health alterations, three interobserver reliability checks were performed throughout the period of data collection. By combining teaching, discussion, and comparison between the observers during a specifically organized observer meeting, we intended to minimize possible observer bias. Also, the intensive observer seminar at the beginning of data collection as well as the comprehensive SOP catalogue were aimed at minimizing observer bias. The agreement between the observers with regard to the compared characteristics was basically quite good to very good. However, the comparisons could not be carried out with the desirable sample size, as either animal welfare aspects (such as too long fixation of the examined animals) or a lack of available animals for the comparison limited the possible sample size at the selected locations. Here, too, it can be seen that the feasibility of such a large study can stand in the way of desirable methodological quality.

Disease prevalences (Table 6): The right-skewed distribution of the prevalence estimates by the farmers indicates that there were either actually some farms with very high prevalences or that some farmers were able to make a more accurate and thus higher estimation of the prevalences than others. We not only collected the estimation of the farmers in the project, but also made a prevalence determination ourselves. Jensen et al. [8] could show that “on average, farmers were conscious of only 45.3% (North), 24.0% (East), and 30.0% (South) of their lame cows”. This fits with the findings of other studies such as Ranjbar et al. [20] who reported a 3.7 times underestimation of lameness in pasture-based herds in Australia. Denis-Robichaud [21] also reported an underestimation of lameness prevalence by farmers and veterinarians. It can be assumed that all farmers’ data on disease prevalence are underestimated—especially if the farmers cannot consult any documentation (such as veterinary invoices or DHI reports, etc.) for estimation. Furthermore, regional differences in disease incidence need not necessarily be explained by differences in husbandry or management. Rather, it can be assumed that the differences in the stringency of disease documentation are mainly due to farm size.

Lameness prevalence in our study was a bit higher than in other studies, as recently reported in a literature review on lameness in dairy cows worldwide and over 30 years with study means ranging from 5.1 to 45% [22]. In Europe, only a few recent reports of mastitis prevalence or reproduction parameters exist. One group carried out a meta-analysis on subclinical ketosis and reported a mean global prevalence of 22.7%, which is much higher than in our study [23].

The differences in farm size between the regions resulted in many structural differences, starting with more part-time farmers and less employees including a lower educational level in the South, and more automated milking systems and IVHM contracts in the East. Grazing was most common in the North because farms have more grassland there. The fact that some diseases had lower incidences in the South although the husbandry conditions were not better (e.g., tethering stalls) underpins the multifactorial character of the diseases that were investigated.

In conclusion, we succeeded in having a representative sampling, as much as possible, in terms of herd size and region. Every study with voluntary participation will not be fully representative, and even if it is possible to control for bias in the data analysis (e.g., if the distribution of the herd size in the target population is known), information about underrepresented strata (e.g., small herds) will remain less reliable. In our study, we were able to stratify the farms by herd size and region because we had access to lists of all dairy herds in Germany. If such population data are not available, convenience sampling may also lead to reliable results, as long as the sources of bias are discussed thoroughly.

Number of lactations: Assuming that a cow gave birth to a calf for the first time at the age of 24–36 months, followed by one calf, i.e., one lactation, approximately, per year, it can be estimated that cows reached an age of about four to six years on average. From an economic standpoint, this is not ideal since the full potential of a cow’s performance is not reached before the sixth lactation, and rearing costs on average are not paid back earlier than in the third lactation [24]. The differences observed between the regions might be associated with the predominant breed in the regions because the Simmental breed was mainly present in region South and is known to reach higher lactation numbers due to less intensive use.

Culling and Mortality: The term culling includes animals exiting the farm due to slaughter or sale [25]. The reported numbers for culling risk in this study resemble those found in Germany during the last decades without clear changes over the years (BRS 2023). In a meta-analysis by Compton et al. [26], including data from studies in North America, Europe (except Germany) and Australia/New Zealand, lower culling risks were reported (incidence risk from 0.14 to 0.28). The authors also did not find any evidence of overall change in culling incidence risk over time. With regard to mortality, a threshold of 2% has been recommended [27]. In regions North and East, this threshold was exceeded by at least every second farm. In the South, this was the case for at least every fourth farm. Our figures for mortality rate are in alignment with an earlier study in Germany, where an overall mortality rate of 0.047 per animal-year was reported [28], and other international studies [26]. Due to our study design with the limitation of the observation period to one year, no inference about temporal changes could be drawn. However, in the literature, an increase in mortality has been described [26]. When looking at the reasons as to why animals leave the farm, our data revealed that roughly 50% were due to disease. Due to the fact that two different databases had to be consulted for the analysis of culling and mortality (HIT) and culling reasons (DHI), respectively, there is no information as to why animals die on the farm. A study by Alvåsen et al. [29] showed that the main reasons for mortality and slaughtering differed; e.g., more cows with fertility disorders or udder diseases were slaughtered, whereas metabolic diseases and claw/leg disorders more often were reported for mortality disposal. Accurate documentation, on a farm level, produces information that might be helpful for further disease management.

### 4.4. Risk Factor Models

Selection and confounding bias need to be considered in observational studies [17]. The way to deal with it is to detect and control it. This is best carried out by drawing causal directional acyclic graphs (causal DAGs) that include not only the influence factor and the outcome of interest, but also all factors that might have led to biased selection and that are a common cause of the influence factor and the outcome. These types of bias can be controlled, e.g., by stratification [30].

As described above, the amount of regression models in the final report was too large to give each of the models the time and effort needed for this well-founded confounder analysis. We needed a more pragmatic approach to be able to finish the report on time. Proper confounder analyses are planned for future scientific publications.

Statistical analysis: Although this was not at all foreseen in the sample design, it was still possible to create risk models with little, describable (and thus controlled) bias. It has also to be stated that the sample size in some cases was not large enough to satisfy the requirements of a risk factor analysis, neither for univariable nor much less for multivariable models.

Our pragmatic approach for the final report followed a standardized and validated procedure that is described above. This procedure was developed and agreed on by all study personnel that conducted regression models. This avoided “statistician-bias” and ensured the highest possible quality and thus can be regarded as a good compromise between scientific demands and resources.

The risk factor models allowed insights into causal associations concerning, e.g., lameness [31], udder health [32], and infectious diseases [33], but also concerning calf and young cattle health [8]. Interestingly, differences could be seen between regions in most cases. These can be used as starting points for future investigations of specific topics.

## 5. Conclusions

The perfect epidemiologic study does not exist. Non-biased results can only be obtained from experimental studies, and these lack the variance that exists in real life. Therefore, the best that can be achieved is to carry out observational studies that are as representative and as free from bias as possible to record all known sources of confounding and to analyze then and interpret the data as carefully as possible. Not only in large studies but in all, proper project management and comprehensive team communication are necessary to identify and solve all the challenges of a technical or methodic nature that may occur during a study. In the case of the PraeRi study, we have almost representative estimates of the most important animal health disorders in dairy husbandry in Germany. These, in parts alarmingly high, prevalences and incidences, e.g., for lameness, can and have already been used as a basis for in-depth analyses of certain aspects and can contribute a lot to increasing dairy health and welfare.

## Figures and Tables

**Figure 1 animals-14-01385-f001:**
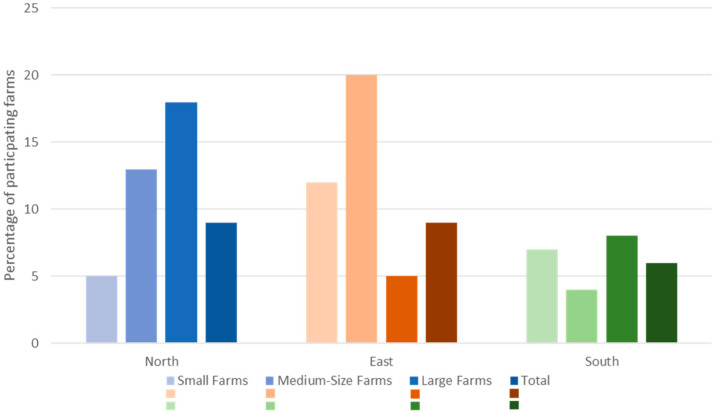
Overview of participation rate by region and farm size category in a prevalence study among 765 German dairy herds. Displayed is the percentage of farms that participated from all farms that were invited.

**Figure 2 animals-14-01385-f002:**
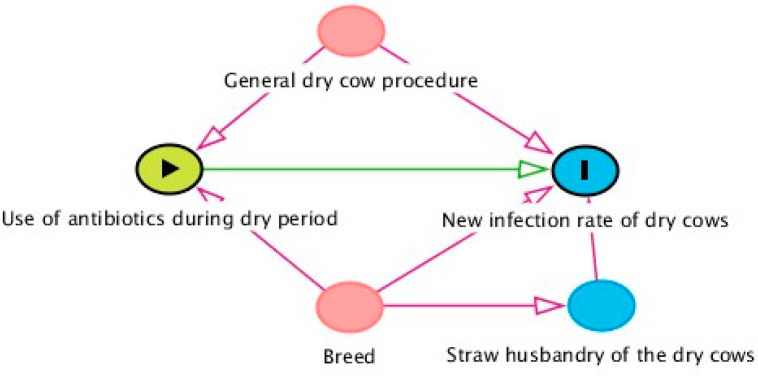
A causal diagram displayed as a directed acyclic graph in a prevalence study among 765 German dairy herds showing the example target variable “New infection rate of dry cows”, its influencing variable “Use of antibiotics during the dry period”, and the possible confounders.

**Figure 3 animals-14-01385-f003:**
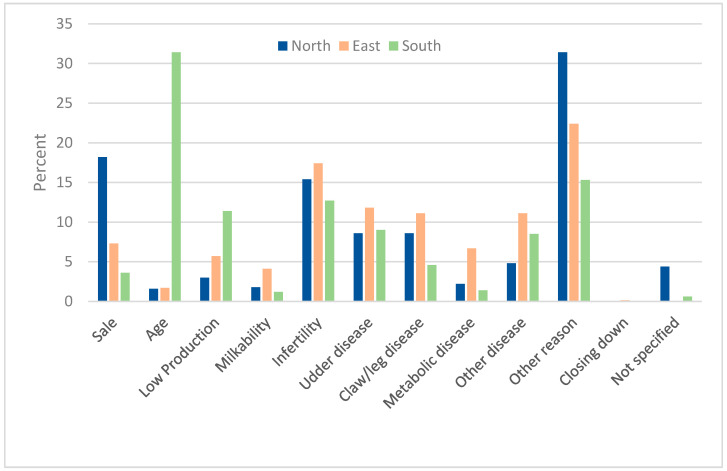
Reasons for cows leaving the farm due to sale or culling in a prevalence study among 765 German dairy herds. Shown is the mean proportion of cows (in% of cows that left) that left the farm for the reasons listed, staggered by region: North (n = 242 farms), East (n = 249 farms), and South (n = 232) farms (database: DHI).

**Table 1 animals-14-01385-t001:** Adapted cut-offs (number of cows) based on data originating from state control associations as opposed to data from the National Traceability and Information System for Animals in a prevalence study among 765 German dairy herds (data as of September 2016).

	Cut-Off		
Region	Small	Medium	Large
North	1–64	65–113	≥114
East	1–160	161–373	≥374
South	1–29	30–52	≥53

**Table 2 animals-14-01385-t002:** Overview of participation rate by region and farm size category in a prevalence study among 765 German dairy herds.

	Number of Small Farms		Number of Medium-Sized Farms		Number of Large Farms		Total	
	Invited	Visits	Invited	Visits	Invited	Visits	Invited	Visits
Schleswig- Holstein	330	13	210	31	110	25	650	69
Lower Saxony	1334	70	464	59	339	55	2137	184
North	1664	83/5%	674	90/13%	449	80/18%	2787	253/9%
Mecklenburg- Western Pomerania	189	18	123	26	264	22	576	66
Saxony-Anhalt	156	20	131	26	82	26	369	72
Brandenburg	109	24	103	22	173	19	385	65
Thuringia	247	20	76	13	86	16	409	49
East	701	82/12%	433	87/20%	605	83/5%	1739	252/9%
Bavaria	1345	92	2015	84	1058	84	4418	260
South	1345	92/7%	2015	84/4%	1058	84/8%	4418	260/6%

**Table 3 animals-14-01385-t003:** Number of animals examined in a prevalence study including 765 German dairy herds.

	Number of Examined Calves		Number of Examined Cows		Number of Examined Young Stock	
	Average/ Farm	Total	Average/ Farm	Total	Average/Farm	Total
North	15	3741	99	24,980	77	19,571
East	37	9188	198	49,936	222	56,058
South	8	2074	44	11,388	35	9224
Total	20	15,003	113	86,304	111	84,853

**Table 4 animals-14-01385-t004:** Representativeness of the PraeRi study population compared with data from the stat. Bundesamt (Destatis/GENESIS database, status as of 1.3.2016) (PraeRi: prevalence study among 765 German dairy herds).

	Number of Farms		Number of Cows		% of Farms in Study	% of Cows in Study	Average Farm Size		Distribution of Farms in Region (%)		Distribution of Cows in Region (%)	
	Genesis	PraeRi	Genesis	PraeRi	PraeRi	PraeRi	Genesis	PraeRi	Genesis	PraeRi	Genesis	PraeRi
Lower Saxony	10,080	184	864,750	18,188	2	2	86	99	71	73	69	70
Schleswig- Holstein	4180	69	396,358	7829	2	2	95	113	29	27	31	30
North	14,260	253	1,261,108	26,017	2	2	88	103				
Brandenburg	539	66	159,964	24,884	12	16	297	377	24	26	28	29
Mecklenburg-Western Pomerania	712	72	180,918	25,504	10	14	254	354	32	29	31	29
Saxony-Anhalt	520	65	123,405	22,352	13	18	237	344	23	26	21	26
Thuringia	485	49	110,502	14,098	10	13	228	288	21	19	19	16
East	2256	252	574,789	86,838	11	15	255	345				
Bavaria	32,564	260	1,208,640	11,539	1	1	37	44	100	100	100	100
South	32,564	260	1,208,640	11,539	1	1	37	44				

**Table 5 animals-14-01385-t005:** Agricultural business branches besides dairying in a prevalence study among 765 German dairy herds.

	Region					
Other Cattle Holdings	North		East		South	
	n	%	n	%	n	%
Rearing calves	242	95.7	249	98.8	255	98.1
Breeding young cattle	239	94.8	237	94.0	243	91.9
Stud bulls	90	35.6	94	37.3	26	10.0
Bull fattening	79	31.2	57	22.6	27	10.5
Heifer fattening	23	9.1	19	7.5	13	5.0
Calf fattening	6	2.4	20	7.9	12	4.6
Suckler cow husbandry	9	3.6	51	20.2	3	1.2
Rearing calves from other farms of origin	29	11.5	13	5.2	10	3.8
Rearing young cattle from other farms of origin	26	10.3	16	6.3	10	3.8
No answer selected/does not apply	11	4.4	4	1.6	0	0.0
Total number of farms	253		252		260	

**Table 6 animals-14-01385-t006:** Mean incidence (%) of selected diseases in cows (based on farmers’ interview data) in a prevalence study among 765 German dairy herds.

Variable	Region	Number of Farms	Mean	Standard Deviation	Me-Dian	25%- Quantile	75%- Quantile	Missing Values
Milk fever	North	250	10.6	10.0	7.4	4.3	14.5	2
	East	240	6.7	7.2	5.0	2.3	9.5	12
	South	257	5.8	5.7	4.7	1.7	8.3	3
Retained placenta	North	251	11.4	8.8	9.7	5.8	14.8	1
	East	236	10.2	8.1	9.2	5.0	13.3	5
	South	258	8.2	5.9	7.3	4.5	11.0	1
Uterine inflammation	North	252	8.8	10.3	5.4	2.6	11.1	1
	East	235	12.0	12.9	8.2	2.9	16.3	17
	South	257	4.5	6.0	3.3	0.0	6.7	3
Pneumonia	North	252	1.5	3.1	0.0	0.0	1.7	1
	East	239	2.3	6.1	0.8	0.0	2.0	13
	South	260	0.7	2.0	0.0	0.0	0.0	0
Ketosis	North	247	7.1	8.1	5.0	2.0	10.0	6
	East	234	6.3	8.3	3.0	1.0	9.0	18
	South	254	2.8	4.8	0.0	0.0	3.9	6
Displaced abomasum	North	253	2.1	2.2	1.6	0.0	3.0	0
	East	243	1.8	2.3	1.0	0.3	2.5	9
	South	260	0.3	1.1	0.0	0.0	0.0	0
Foreign body disease	North	251	1.4	2.5	0.0	0.0	2.0	2
	East	232	1.2	3.4	0.0	0.0	1.0	20
	South	259	0.9	3.2	0.0	0.0	0.0	1
Mastitis without general disorder	North	252	16.3	11.3	14.8	8.7	21.5	1
	East	232	22.0	19.9	15.9	5.9	32.4	20
	South	259	14.2	12.5	11.8	6.0	19.2	1
Mastitis with general disorder	North	252	5.0	5.1	4.0	1.4	6.5	1
	East	228	4.9	8.0	2.0	1.0	5.0	24
	South	260	4.6	7.3	2.6	0.0	6.6	0
Heifer mastitis	North	253	3.4	4.6	2.1	0.0	4.4	0
	East	219	6.3	8.3	3.4	0.9	8.3	33
	South	225	2.5	5.6	0.0	0.0	3.3	5
Abortion	North	250	2.8	2.5	2.2	1.4	4.0	3
	East	239	2.9	2.9	2.0	1.0	4.0	13
	South	260	2.9	3.3	2.5	0.0	4.1	0
Lameness	North	252	12.6	12.1	9.5	4.8	16.4	1
	East	250	13.8	14.9	9.5	3.8	20.0	2
	South	259	8.5	7.8	7.1	3.3	11.5	1

## Data Availability

The data sets presented in this article are not readily available because the data were acquired through cooperation between different universities. Therefore, any data transfer to interested persons is not allowed without an additional formal contract. Data are available for qualified researchers who sign a contract with the project consortium. This contract will include guarantees of the obligation to maintain data confidentiality in accordance with the provisions of German data protection law. Currently, there exists no data access committee nor another body who can be contacted for the data; a committee will be founded for this purpose. This future committee will consist of the authors as well as members of the related universities. Interested cooperative partners, who are able to sign a contract as described above, may contact MH, Clinic for Cattle at the University of Veterinary Medicine, Hannover, Bischofsholer Damm 15, 30173 Hannover, Germany, Email: martina.hoedemaker@tiho-hannover.de. Requests to access the data sets should be directed to martina.hoedemaker@tiho-hannover.de.

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
