# Peer review of "Application of Epidemiological Methods in a Large-Scale Cross-Sectional Study in 765 German Dairy Herds—Lessons Learned"

_animals, 2024, doi:10.3390/ani14091385_

Round 1
Reviewer 1 Report
Comments and Suggestions for Authors
Very well structured article. Congratulations!
Some points:
- Title should be improved (to better clarify that dairy herds were evaluated)
- Some typping errors (need to check)
- Colors of bars from Figure 1 are confuse, because the blue color of the subtitle is the same color for North region and different for the other studied regions
- I didn't find the information regarding Ethical Comittee Approval for the study
- I didn't find the information about the signature of the "Free and informed consent form" by farmers who participated in the study
Comments on the Quality of English Language
There are some typping errors that must be corrected.
Author Response
Some points:
- Title should be improved (to better clarify that dairy herds were evaluated)
Dear reviewer, thank you for this comment. We changed the title as follows:
Application of epidemiological methods in a large-scale cross-sectional study in 765 Germany dairy herds – lessons learned
- Some typping errors (need to check)
Thank you, we carefully revised the manuscript concerning English language and typos.
- Colors of bars from Figure 1 are confuse, because the blue color of the subtitle is the same color for North region and different for the other studied regions
We added the colors of South and East
- I didn't find the information regarding Ethical Comittee Approval for the study
Dear reviewer, the following ethical statements was sent to the editorial office on March, 25th, 2024 and now included in the manuscript in ll. 998 - 1003: “Ethical review and approval were waived for the study on human participants in accordance with the local legislation and institutional requirements in 2015. The participants provided their written informed consent to participate in this study. Ethical review and approval was not required for the animal study because no painful interventions have been made. This was in accordance with the local legislation and institutional requirements. Written informed consent was obtained from the owners for the participation of their animals in this study.”
- I didn't find the information about the signature of the "Free and informed consent form" by farmers who participated in the study.
This document was also sent to the editorial office on March, 25th, 2024.
Reviewer 2 Report
Comments and Suggestions for Authors
The objective of the study was to present the problems that can be encountered during large-scale field studies under real conditions.
The authors present with frankness and open-mindness the potential problems that can be seen in large-scale studies. I believe that these issues must be described, so that reviewers and editors in scientific journals stop asking for tasks that are really impossible to fulfil.
So, I support publication, but I have some points that need to be attended.
1. The authors should add a sub-section about the lack of controls in such studies. Frequently, reviewers and editors make a fuss regarding lack of controls and this must be brought to their attention.
2. The authors should add a sub-section to compare about the advantages of using convenient inclusion of farms into field studies. In reality, convenient sampling is the only the way to perform large-scale field studies. The authors must provide their personal view: if they did the work again, would they carry out convenience inclusion?
3. Comment about tables and figures. These are very ok, the visualization of results is fine.
4. References. Please provide a comparison with other relevant findings in other countries.
5. The Conclusions are not fully consistent with the findings. Please rewrite and please tone down the conclusions to be in line with the findings.
Overall. The manuscript requires a few changes before resubmission to improve it before final acceptance. A manuscript that merits publication.
Recommendation. Accept after minor revision as suggested.
Author Response
The objective of the study was to present the problems that can be encountered during large-scale field studies under real conditions.
The authors present with frankness and open-mindness the potential problems that can be seen in large-scale studies. I believe that these issues must be described, so that reviewers and editors in scientific journals stop asking for tasks that are really impossible to fulfil.
So, I support publication, but I have some points that need to be attended.
- The authors should add a sub-section about the lack of controls in such studies. Frequently, reviewers and editors make a fuss regarding lack of controls and this must be brought to their attention.
Dear reviewer, thank you for this idea. We added the following in ll. 823-827: “This study is an epidemiological study, not an experiment. The study type is referred to as a cross-sectional study and follows specific rules for planning, implementation and evaluation. It differs from other epidemiological studies in that, among other things, no control group is used here, as no scientifically selected groups are compared, but the status quo in the existing population is surveyed.”
- The authors should add a sub-section to compare about the advantages of using convenient inclusion of farms into field studies. In reality, convenient sampling is the only the way to perform large-scale field studies. The authors must provide their personal view: if they did the work again, would they carry out convenience inclusion?
Dear reviewer, thank you for this important comment. We added the following in the section “4.3. Prevalence estimation” of the discussion (ll. 878-886): “In conclusion, we succeeded to have a representative sampling as good as possible in terms of herd size and region. Every study with voluntary participation will not be fully representative, and even if it is possible to control for bias in the data analysis (e.g. if the distribution of the herd size in the target population is known), information about underrepresented strata (e.g. small herds) will remain less reliable. In our study, we were able to stratify the farms by herd size and region, because we had access to lists of all dairy herds in Germany. If such population data are not available, convenience sampling may also lead to reliable results as long as the sources of bias are discussed thoroughly.”
- Comment about tables and figures. These are very ok, the visualization of results is fine.
Thank you
- Please provide a comparison with other relevant findings in other countries.
Dear reviewer, thank you for this comment. We added some references in the discussion, i.e. in ll. 850 – 870.
- The Conclusions are not fully consistent with the findings. Please rewrite and please tone down the conclusions to be in line with the findings.
Thank you, we overworked the conclusions to be more general as follows: “The perfect epidemiologic study does not exist. Non-biased results can only be obtained from experimental studies, and these lack the variance that exists in real life. Therefore, the best that can be done is to carry out observational studies as representative and as free from bias as possible to record all known sources of confounding and to analyze then and interpret the data as carefully as possible. Not only in large studies, proper project management and comprehensive team communication is necessary to identify and solve all the challenges of technical or methodic nature that occur during the study. In the case of the PraeRi study, we have almost representative estimates of the most important animal health disorders in dairy husbandry in Germany. These in parts alarmingly high prevalences and incidences, e.g. for lameness, can and have already been used as basis for in-depth analyses of certain aspects and can contribute a lot to increase dairy health and welfare.”